# Deep Mutational Scanning to Predict Escape from Bebtelovimab in SARS-CoV-2 Omicron Subvariants

**DOI:** 10.3390/vaccines11030711

**Published:** 2023-03-22

**Authors:** Mellissa C. Alcantara, Yusuke Higuchi, Yuhei Kirita, Satoaki Matoba, Atsushi Hoshino

**Affiliations:** 1Department of Cardiovascular Medicine, Graduate School of Medical Science, Kyoto Prefectural University of Medicine, Kyoto 602-8566, Japan; 2Department of Nephrology, Graduate School of Medical Science, Kyoto Prefectural University of Medicine, Kyoto 602-8566, Japan

**Keywords:** deep mutational scanning, escape mutation, bebtelovimab, BA.2

## Abstract

The major concern with COVID-19 therapeutic monoclonal antibodies is the loss of efficacy against continuously emerging variants of SARS-CoV-2. To predict antibody efficacy against future Omicron subvariants, we conducted deep mutational scanning (DMS) encompassing all single mutations of the receptor-binding domain of the BA.2 strain utilizing an inverted infection assay with an ACE2-harboring virus and library spike-expressing cells. In the case of bebtelovimab, which preserves neutralization activity against BA.2 and BA.5, a broad range of amino acid substitutions at K444, V445, and G446, and some substitutions at P499 and T500, were indicated to achieve the antibody escape. Among subvariants with current rises in case numbers, BA2.75 with G446S partially evaded neutralization by bebtelovimab, while complete evasion was observed in XBB with V445P and BQ.1 with K444T. This is consistent with the DMS results against BA.2, highlighting the potential of DMS as a predictive tool for antibody escape.

## 1. Introduction

The Omicron variant has demonstrated a tremendous capability to escape from neutralization by vaccinated and convalescent serum samples and clinically used antibody drugs [1,2,3]. Omicron subvariants continuously acquire mutations in the receptor-binding domain (RBD), and the accumulated mutations have provided resistance against the majority of therapeutic antibodies. As of September 2022, among all clinically used therapeutic antibodies, only bebtelovimab has retained efficacy against the major subvariants BA.2 and BA.5 [4,5,6]. 

To predict the neutralization activity against future variants, we performed deep mutational scanning (DMS), with the library encompassing all single amino acid substitutions in the BA.2 RBD. The library was expressed in the context of the full-length spike on the cell surface, and antibody neutralization was reproduced in an inverted infection assay wherein ACE-harboring viruses infected spike-expressing cells [1]. This DMS identified K444, V445, and G446 as the critical sites for the effective neutralization of bebtelovimab. Consistently with this result, two subvariants which are currently showing rises in case numbers, XBB and BQ.1 [7], are resistant to bebtelovimab due to mutations in V445P and K444T, respectively.

## 2. Methods

### 2.1. Cell Culture

HEK 293T cells (#632180, Takara Bio, Kusatsu, Japan) and their derivative, 293T/ACE2 (GenBank accession number NM_001371415) cells, were cultured at 37 °C with 5% CO_2_ in Dulbecco’s modified Eagle’s medium (DMEM, WAKO) containing 10% fetal bovine serum (Gibco, Waltham, MA, USA) and penicillin/streptomycin (100 U/mL, Thermo Fisher Scientific, Waltham, MA, USA). Expi293F cells (#A14527, Thermo Fisher Scientific, Waltham, MA, USA) were grown in Expi293 Expression Medium (Thermo Fisher Scientific, Waltham, MA, USA) containing penicillin/streptomycin (100 U/mL, Thermo Fisher Scientific, Waltham, MA, USA) at 130 rpm, 8% CO_2_, 37 °C. All cell lines routinely tested negative for mycoplasma contamination. 

### 2.2. Protein Synthesis and Purification

The sequence of bebtelovimab was obtained from KEGG DRUG Database “https://www.genome.jp/kegg/drug/ (accessed on 1 February 2022)” and formulated in the form of human IgG1/kappa by using synthetic DNA coding for the variable regions of heavy and light chains taken from the publicly available amino acid sequences, and recombinantly produced in Expi293F cell expression system (Thermo Fisher Scientific, Waltham, MA, USA) according to the manufacturer’s protocol. Fc-fusion proteins were purified from conditioned media using the rProtein A Sepharose Fast Flow (Cytiva, Tokyo, Japan). Fractions containing target proteins were pooled and dialyzed against phosphate buffered saline (PBS).

### 2.3. Deep Mutational Scanning for Escape from Monoclonal Antibodies

Monoclonal antibody selection experiments were performed in biological duplicate using a deep mutational scanning approach with previously described mutant RBD libraries [1]. The library was focused on the BA.2 strain spike RBD region from F329 to C538. Pooled oligos with degenerate NNK codons were synthesized by Integrated DNA Technologies, Inc. (Coralville, IA, USA) Synthesized oligos were extended by overlap PCR and cloned into pcDNA4TO HMM38-HA full-length spike plasmids. Transient transfection conditions were used that typically provide no more than a single coding variant per cell [8]. Expi293F cells at 2 × 10^6^ cells per mL were transfected with a mixture of 1 ng of library plasmid with 1 μg of pMSCV empty plasmid per mL using ExpiFectamine (Thermo Fisher Scientific, Waltham, MA, USA). Twenty-four hours after transfection, cells were incubated with human ACE2 (hACE2)-harboring green fluorescent protein (GFP) reporter viruses, which were generated by transfecting pcDNA4TO hACE2, psPAX2 (addgene #12260), and pLenti GFP into LentiX-293T cells with Lipofectamine 3000 (Thermo Fisher Scientific, Waltham, MA, USA). The viruses that carry hACE2 instead of glycoprotein can infect cells expressing the spike. To analyze the escape mutations from antibodies, cells were pre-incubated with antibodies for 1 h. Next, these cells were treated with hACE2-harboring virus for 1 h and further incubated with fresh medium for 24 h. Cells were harvested and washed twice with PBS containing 10% bovine serum albumin (BSA) and then co-stained for 20 min with anti-hemagglutinin (HA) Alexa Fluor 647 (clone TANA2, MBL, Tokyo, Japan). Cells were again washed twice before being sorted on a MA900 cell sorter (Sony, Tokyo, Japan). Dead cells, doublets, and debris were excluded by first gating on the main population by forward and side scatter. From the HA positive (Alexa Fluor 647 positive) population, GFP-positive and -negative cells were collected. The total numbers of collected cells were about 2 million cells for each group. Total RNA was extracted from collected cells using TRIzol (Thermo Fisher Scientific, Waltham, MA, USA) and Direct-zol RNA MiniPrep (Zymo Research, Irvine, USA) according to the manufacturer’s protocol. First-strand complementary DNA (cDNA) was synthesized with PrimeScript II Reverse Transcriptase (Takara Bio, Kusatsu, Japan) primed with a gene-specific oligonucleotide. The RBD region was divided into 3 sections with 210 bp fragment each and DMS libraries were separately generated for each section and pooled before transfection to analyze the same experimental condition. After cDNA synthesis, each library was amplified with specific primers for each section. Following a second round of PCR, primers added adapters for annealing to the Illumina flow cell, together with barcodes for each sample identification. The PCR products were sequenced on an Illumina NovaSeq 6000 using a 2 × 150 nucleotide paired-end protocol in the Department of Infection Metagenomics, Research Institute for Microbial Diseases, Osaka University. Data were analyzed comparing the read counts of infected GFP-positive population and uninfected GFP-negative one. Log10 enrichment ratios for all the individual mutations were calculated and normalized by subtracting the log10 enrichment ratio for the wild-type sequence across the same PCR-amplified fragment.

### 2.4. Pseudotyped Virus Neutralization Assay

Pseudotyped reporter virus assays were conducted as previously described [1,9]. Using a plasmid encoding the SARS-CoV-2 spike protein (addgene #145032) as a template, D614G mutant and Omicron subvariants were cloned into pcDNA4TO (Thermo Fisher Scientific, Waltham, MA, USA) in the context of the ΔC19 (19 amino acids deleted from the C terminus) [10]. Spike protein-expressing pseudoviruses with a luciferase reporter gene were prepared by transfecting plasmids (pcDNA4TO Spike-ΔC19, psPAX2 (addgene #12260), and pLenti firefly) into LentiX-293T cells with Lipofectamine 3000 (Thermo Fisher Scientific, Waltham, MA, USA). After 48 h, supernatants were harvested, filtered with a 0.45 μm low protein-binding filter (SFCA), and frozen at −80 °C. The 293T/ACE2 cells were seeded at 10,000 cells per well in 96-well plates. Pseudoviruses and three-fold dilution series of antibodies were incubated for 1 h, and these mixtures were then added to 293T/ACE2 cells. After 1 h incubation, the medium was changed. At 48 h post-infection, cellular expression of the luciferase reporter, indicating viral infection, was determined using ONE-Glo Luciferase Assay System (Promega, Madison, WI, USA). Luminescence was read on Infinite F200 pro system (Tecan, Maennedorf, Switzerland). This assay was performed in four replicates and the 50%-neutralizing concentration was calculated using Prism version 9 (GraphPad Software, San Diego, CA, USA).

### 2.5. Statistical Analysis

Pseudovirus neutralization measurements were conducted in four replicates and relative luciferase units were converted to percent neutralization and plotted with a non-linear regression model to determine 50% neutralizing concentration using GraphPad Prism software (version 9.0.0). 

## 3. Results 

### 3.1. Deep Mutational Scanning for Escape from Bebtelovimab in BA.2

We performed deep mutational scanning (DMS) to determine how all single amino acid substitutions impact the susceptibility of BA.2 to bebtelovimab, the clinically used drug which retains neutralization activity against BA.2 and BA.5 [4,5,6]. The mutant library encompassed all 20 single amino acid substitutions in the RBD (F329 to C538) of the BA.2 spike (Figure 1a). The plasmid library was expressed in human Expi293F cells in a manner wherein cells would typically acquire no more than a single variant [1] (PMID: 29678950, 30723160, 33597251). Cells were pre-incubated with bebtelovimab and subsequently infected with an ACE2-harboring green fluorescent protein (GFP) reporter virus. The presence of a mutant spike that was able to escape the antibody allowed infection by the ACE2-harboring virus, and these cells obtained GFP fluorescence. Meanwhile, mutations with preserved susceptibility to the antibody or compromised infectivity prevented the infection. Both infected GFP-positive cells and control GFP-negative cells were harvested with fluorescence-activated cell sorting (FACS). The RNA was extracted and underwent deep mutational sequencing (Figure 1b). The DMS library was divided into three units and pooled after plasmid construction. Sequence samples were prepared with primers specific to each unit and mutations were identified by the direct sequencing of the whole unit. The escape value was defined as the ratio of GFP-positive read count to the GFP-negative read count and normalized by the value of the wild-type [1]. The spike protein-expressing cells constituted 20–30% of transfected cells, which was suitable to avoid multiple library induction [11]. ACE2-harboring virus infected ~15% of transfected cells, and bebtelovimab was titrated to reduce the infection rate to ~3% [1] (Figure 1c). DMS experiments were performed in duplicate and exhibited good reproducibility, R^2^ (coefficient of determination) = ~0.4796, as was the case in our previous and other studies (Figure 1d) [1,12,13]. 

The BA.2-based DMS indicated that a broad range of amino acid substitutions at K444, V445, and G446, and some substitutions at P499 and T500, contribute to the escape from bebtelovimab, which is consistent with previous reports [14,15] (Figure 1e). These sites were fairly unchanged before the emergence of the Omicron variant; however, Omicron subvariants later obtained a wider variety of mutations, including substitutions at K444, V445, and G446.

### 3.2. XBB and BQ.1 Evade Bebtelovimab Neutralization

We then evaluated the neutralization activity of bebtelovimab against major Omicron subvariants (Figure 2a). In the neutralization assay using pseudotyped lentiviruses, BA.2, BA.2.12.1, BA.4/5, and BA.4.6 were susceptible to bebtelovimab. On the other hand, BA.2.75 obtained modest resistance, while XBB and BQ.1 obtained dramatic resistance against bebtelovimab (Figure 2b). Another study also reported that bebtelovimab had lower neutralization activity against BA.2.75, albeit its activity against BA.1, BA.2, and BA.4/5 was preserved [17]. BA.2.75, a BA.2-derived subvariant, additionally obtained D339H, G446S, and N460K in the RBD, and the result of the DMS indicated that G446S was responsible for reduced neutralization sensitivity against bebtelovimab. XBB and BQ.1 carry V445P and K444T, respectively, and these mutations were indicated in the DMS to result in the strong escape. Although K444, V455, and G446 were regions responsible for bebtelovimab neutralization, not all amino acid substitutions provided escapability. In particular, V445I, V445L, and G446D were observed among circulating SARS-CoV-2 isolates and maintained sensitivity against bebtelovimab (Figure 1e).

## 4. Discussion

After the emergence of SARS-CoV-2, therapeutic monoclonal antibodies (mAbs) were rapidly developed and clinically used to prevent both infection and disease progression [16]. For COVID-19, virus-neutralizing mAbs became the popular anti-viral strategy. However, the mutational escape of the virus from monoclonal antibodies was observed since the earliest periods of their clinical use. Bamlanivimab was the first mAb to be granted an Emergency Use Authorization (EUA) by the U.S. Food and Drug Administration (FDA) to treat COVID-19 in November 2020, but the spread of E484K-carrying variants led to its withdrawal in April 2021 [18]. To prevent such viral evasion, some mAbs were used in cocktail form, and others were designed to target highly conserved regions [14,19,20]. However, considerable antibody escape was observed in the Omicron variants and continuously emerging subvariants continue to threaten the efficacy of therapeutic mAbs. Thus, it is crucial to comprehensively understand antibody escape to effectively manage future variants. 

DMS is a powerful tool to understand functional alteration caused by each amino acid substitution and can be applied to predict antibody escape. One approach used to evaluate antibody escape is the assay for antibody binding using the yeast-surface display of the RBD domain. This method requires the separate acquisition of data on the alteration of ACE2-binding to assess the viral infectivity [21,22]. Instead of using this method, we developed an original assay system that would reproduce the viral infection. The full-length spike protein was expressed in human Expi293F cells and incubated with ACE2-harboring pseudotyped lentivirus. This assay mimics the infection well: the fusion of cell and viral membranes is mediated by protein–protein interactions between the virus spike and ACE2, albeit in an inverted orientation. When examined with antibodies, this assay can directly assess the viral escape that results from antibody-binding and infectivity.

Crystal structure analyses provide information regarding the antibody epitope; however, not all epitope sites, and not all amino acid substitutions, are responsible for antibody escape (Figure 1e). DMS can comprehensively identify the antibody escape, and this detailed characterization leads to the effective management of future SARS-CoV-2 variants.

## Figures and Tables

**Figure 1 vaccines-11-00711-f001:**
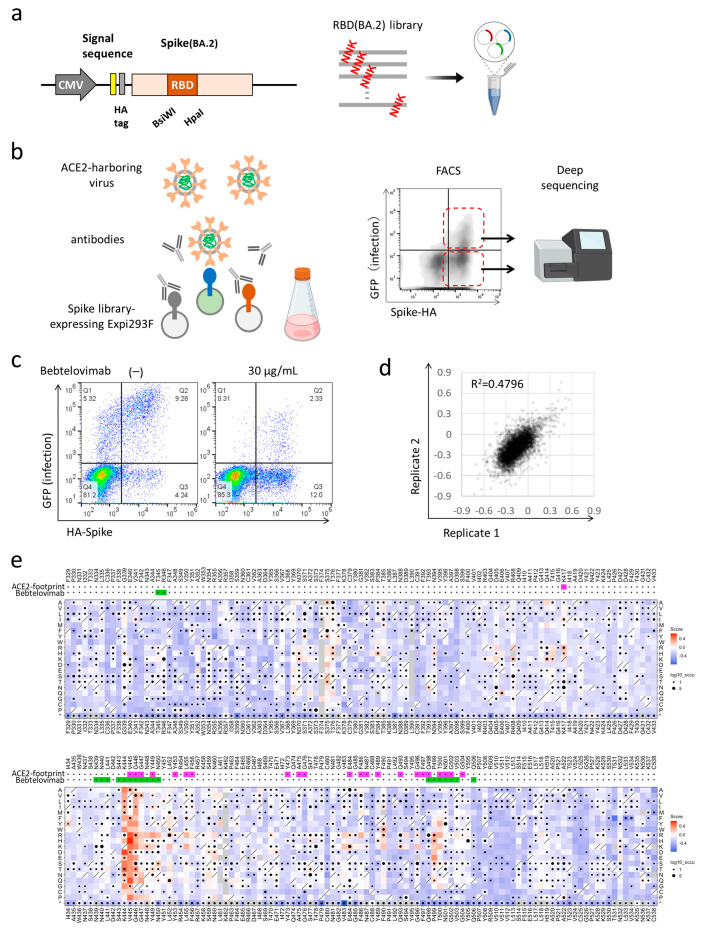
Deep mutational scanning identified single-residue mutation to induce escape from bebtelovimab. (**a**) A design of the deep mutational scanning (DMS) library for the receptor-binding domain of BA.2 strain. (**b**) A schematic of the DMS approach to evaluate escape from neutralizing antibodies in the inverted infection assay; FACS, fluorescence-activated cell sorting. Red dotted line indicates populations sorted for GFP-positive infected and GFP-negative un infected cells. (**c**) The proportion of library-expressing and and pseudovirus-infected cells. Q1 and Q2 are population infected by pseudovirus. Q2 and Q3 are population expressing library spike protein. (**d**) The correlation of mutation effects on the alteration of escape from bebtelovimab across replicates. (**e**) The heatmap illustrating how all single mutations affect the escape from bebtelovimab. Squares are colored by mutational effect according to scale bars on the right, with blue indicating deleterious mutations. Squares with a diagonal line through them indicate BA.2 strain amino acid. Black dot size reflects the frequency in the virus genome sequence according to GISAID database as of 17 September 2022. ACE2 and bebtelovimab footprints are highlighted in magenta and light green, respectively. Footprints on RBDs were defined according to 5 Å distance from ACE2- or monoclonal antibody-contacting residues [16]. Some figures are adapted from images created with BioRender.com.

**Figure 2 vaccines-11-00711-f002:**
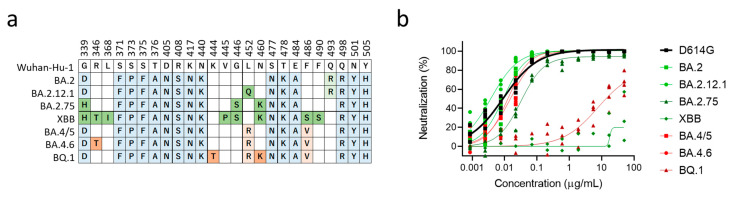
Bebtelovimab neutralization efficacy against the Omicron subvariants. (**a**) Location of mutations in the receptor-binding domain proteins of the Omicron subvariants: BA.2 and its three derivatives, BA.2.12.1, BA.2.75, XBB, and BA.4/5, and its two other derivatives, BA.4.6 and BQ.1.1. Conserved mutations among Omicron subvariants are highlighted in blue. Unique mutations in BA.2 derivatives and BA.4/5 derivatives are highlighted in green and red, respectively. Top number is the residue according to the spike protein of SARS-CoV-2 Wuhan-Hu-1. Data are adapted from Coronavirus Antiviral & Resistance Database of Stanford University “https://covdb.stanford.edu/ (accessed on 1 November 2022). (**b**) Neutralization efficacy of bebtelovimab in 293T/ACE2 cells. n = 4 technical replicates.

## Data Availability

Deep mutational scan data has been deposited at SpikeDB “https://sysimm.ifrec.osaka-u.ac.jp/sarscov2_dms/ (accessed on 1 December 2022)”.

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
