# Peer review of "Deep Mutational Scanning to Predict Escape from Bebtelovimab in SARS-CoV-2 Omicron Subvariants"

_vaccines, 2023, doi:10.3390/vaccines11030711_

Round 1

Reviewer 1 Report

1.  The title is too generalized.  In this study, DMS was only used to predict antibody escape with respect to bebtelovimab while the title implies DMS was used to predict antibody escape from more than one monoclonal antibody.

2.  English grammar needs to be corrected in some places.  For instance, in line 164, region should be regions.  In lines 155 to 167, sometimes the authors used the present tense and at other times the authors used the past tense.  Also, in lines 189-190 "to effectively manage" is repeated.

Author Response

Thank you for pointing out important issues. As indicated, we have changed the title as "Deep mutational scanning to predict escape from bebtelovimab in SARS-CoV-2 Omicron subvariants", and also this manuscript underwent intensive English revision.

Reviewer 2 Report

The continuing emerged SARS-CoV-2 variants might be the major challenge for the therapeutic monoclonal Nabs. Alcantara et. al tried to employ the saturated mutagenesis approach to investigate and predict the site mutations in the BA.2 spike which would be influential to the efficacy of the Nabs. The data in this manuscript provided a comprehensive and valuable picture of all the potential mutations in the context of BA.2 and their effects on the therapeutics. Prior to acceptance, some questions should be addressed.

1.     It is known that the site mutations in the spike protein could not only change the antigenicity but also the infectivity of the SARS-CoV-2. In this reverted infection assay, could the spikes with the single-site mutations in the target cells alter the infectivity of the reporter viruses? If so, how to separate the lower infectivity yielded by mutations influencing the infectivity or the antibody escapablility. If not, please provide the data for this.

2.     G446S is one of the mutations identified in this manuscript responsible for the reduced neutralization sensitivity. BA.2.75 which have this mutation didn’t show obvious reduction. And the BA.1 with this mutation remained sensitive to Bamlanivimab in the ref 16 line 160. How to explain it? 

3.     The identified mutations in DMS are very valuable for evaluation of the Bamlanivimab efficacy. It is recommended to validate the findings in DMS at least using the PBNA.

4.     Some descriptions in the method section is not clear and not properly referred, which make it not easy to follow the process of the experiment and analysis. In Line 119, how to ensure the single variant is not described in the ref 1. Pls provide the original ref with how and why for this step.

5.     In Line 126, how to divide the library into 3 units. Pls give some details and why.

6.     For Figure 1e, how to make the figure, pls give the details. 

7.     In line 152, how to get the footprint for the ACE2 and the antibody by experiment or reference. If the latter, pls give the citation.

Author Response

Rev#2

The continuing emerged SARS-CoV-2 variants might be the major challenge for the therapeutic monoclonal Nabs. Alcantara et. al tried to employ the saturated mutagenesis approach to investigate and predict the site mutations in the BA.2 spike which would be influential to the efficacy of the Nabs. The data in this manuscript provided a comprehensive and valuable picture of all the potential mutations in the context of BA.2 and their effects on the therapeutics. Prior to acceptance, some questions should be addressed.

We thank the reviewer for appreciating the importance of our study and recognizing the potential of our DMS approach as a tool to predict antibody escape.

  1. It is known that the site mutations in the spike protein could not only change the antigenicity but also the infectivity of the SARS-CoV-2. In this reverted infection assay, could the spikes with the single-site mutations in the target cells alter the infectivity of the reporter viruses? If so, how to separate the lower infectivity yielded by mutations influencing the infectivity or the antibody escapablility. If not, please provide the data for this.

Yes, if we conduct this DMS without antibodies, we can assess the alteration of infectivity due to single amino acid substitutions. In this study, we haven’t analyzed infectivity but we previously performed the DMS for infectivity, as well as affinity and expression level in Wuhan strain-based library (Sci Transl Med. 2022 Jun 22;14(650):eabn7737. PMID: 35471044). In the real world, escape mutations are the mutations that lose the affinity to antibody but preserve infectivity. Previous approach needs to analyze both infectivity and affinity to antibodies separately and integrate them to understand the escapability. However, our assay can directly assess the viral escape that results from antibody-binding and infectivity. We discuss this advantage over standard yeast-surface display method in the section of Discussion.

  1. G446S is one of the mutations identified in this manuscript responsible for the reduced neutralization sensitivity. BA.2.75 which have this mutation didn’t show obvious reduction. And the BA.1 with this mutation remained sensitive to Bamlanivimab in the ref 16 line 160. How to explain it?

I presume that Bamlanivimab is the typo for Bebtelovimab. In the DMS, G446S showed modest escapability, it was faint red, not dark red. Consistent with this result, our PBNA exhibited ~3-fold lower neutralization and the paper (ref 16) indicated ~10-fold lower neutralization. In contrast, V445P was much darker red than G446S, and then XBB exhibited almost complete escape.

  1. The identified mutations in DMS are very valuable for evaluation of the Bamlanivimab efficacy. It is recommended to validate the findings in DMS at least using the PBNA.

As suggested, validation is important to ensure the quality of this DMS. We thus evaluated the escapability of Omicron subvariants. I agree that more large scale validation is better, but there is the limitation of resources and we believe current data is acceptable.

  1. Some descriptions in the method section is not clear and not properly referred, which make it not easy to follow the process of the experiment and analysis. In Line 119, how to ensure the single variant is not described in the ref 1. Pls provide the original ref with how and why for this step.

According to the previous studies, transfection of small amount of library plasmid with 1000~2000 folds empty plasmid achieves the almost single plasmid transfection. In general, Yeast screening has advantages in terms of library size and the ability to restrict incorporation of multiple mutants due to the exclusive nature of different plasmids. In contrast, mammalian cell-based screening has a limit in library size and can be contaminated by cells expressing multiple mutants; however, its strength is that it allows analysis of various phenotypes, including virus infection. Additional reference papers have been included (PMID: 29678950, 30723160, 33597251).

  1. In Line 126, how to divide the library into 3 units. Pls give some details and why.

Our library had no barcode and each mutation was identified by direct sequencing with 150bp pair-end mode. To fit this analysis, the RBD region was divided into 3 sections with 210bp fragment each and DMS libraries were separately generated for each section and pooled before transfection to analyze the same experimental condition. After cDNA synthesis, each library was amplified with specific primers for each section.

  1. For Figure 1e, how to make the figure, pls give the details.

Data were analyzed comparing the read counts of infected GFP-positive population and uninfected GFP-negative one. Log10 enrichment ratios for all the individual mutations were calculated and normalized by subtracting the log10 enrichment ratio for the wild type sequence across the same PCR-amplified fragment.

  1. In line 152, how to get the footprint for the ACE2 and the antibody by experiment or reference. If the latter, pls give the citation.

In the figure legend, we explain it as “Footprints on RBDs were defined according to 5A ° distance from ACE2- or monoclonal anti-body-contacting residues (17).“

Reviewer 3 Report

This is an article about DMS that could be used as a tool to predict antibody escape for SARS CoV-2. The authors should address the following issues:

1.       The article lacks key methodological details. With the provided information, other researchers cannot duplicate the experiments.

2.       Figure 2A; source?

3.       Line 24: obtaining not appropriate word. Replace it with acquiring.

Author Response

We thank the reviwer for the careful reading and accurately understanding the content of the paper in great detail. We have revised the method section and others as requested. Figure 2A was adapted from Coronavirus Antiviral & Resistance Database of Stanford University (https://covdb.stanford.edu/) and mentioned in the figure legend.

Round 2

Reviewer 2 Report

The manuscript has been dramatically improved. Some details and key information are recommended to be provided to make it easy to follow. 

1.     Please provide the primers used as supplementary materials in this communication, including primers used for mutagenesis and sequencing.

2.     The RBD region was divided into 3 sections with 210bp fragment each and DMS libraries were separately generated for each section and pooled before transfection to analyze the same experimental condition.”This step is for mutant library construction or deep sequencing? If for sequencing, a transfection step is needed? Three 210bp section for the F329 to C538? The last section is less than 210bp? A little confusing here. If possible, please provide a schematic figure for the sequencing.

3.     A question about the library construction. After the full-length spike with mutations were cloned into the plasmids, is there an amplification step for the plasmids before the transient transfection, or just transfecting the cells with the ligation products? Pls make it clear.

Author Response

We thank the reviewer for suggestion on the information on DMS methodology. However, this paper is not a protocol paper and we think we already provided the essential information.

Regarding the second issue, the F329 to C538 is 210 amino acid and it is 630bp nucleic acid. Then three 210bp section cover the whole RBD.

Reviewer 3 Report

The authors have satisfactorily addressed my concerns.

Author Response

Thank you for your insightful review.